# Recent wetting trend over Taklamakan and Gobi Desert dominated by internal variability

Wenhao Dong [1,2] ✉, Yi Ming[3], Yi Deng[4] & Zhaoyi Shen [5]

The Taklamakan and Gobi Desert (TGD) region has experienced a pronounced increase in summer precipitation, including high-impact extreme events, over recent decades. Despite identifying large-scale circulation changes as a key driver of the wetting trend, understanding the relative contributions of internal variability and external forcings remains limited. Here, we approach this problem by using a hierarchy of numerical simulations, complemented by diverse statistical analysis tools. Our results offer strong evidence that the atmospheric internal variations primarily drive this observed trend. Specifically, recent changes in the North Atlantic Oscillation have redirected the storm track, leading to increased extratropical storms entering TGD and subsequently more precipitation. A clustering analysis further demonstrates that these linkages predominantly operate at the synoptic scale, with larger contributions from large precipitation events. Our analysis highlights the crucial role of internal variability, in addition to anthropogenic forcing, when seeking a comprehensive understanding of future precipitation trends in TGD.

As the prominent Asian deserts, the Taklamakan and Gobi Desert (TGD) region (black rectangle in Fig. 1a) features a classic interior continental climate with scarce precipitation falling predominantly in summer[1,2]. The relatively infertile soil and sparse vegetation cover lead to a fragile ecosystem that is highly susceptible to variations in summer rainfall[3,4]. Local economic activities, such as agriculture and livestock herding, also rely heavily on the availability of summer rainfall[5,6]. Over the past several decades, the summer rainfall has increased significantly over TGD[7,8]. The wetting trend attracted considerable interest from both the general public and the scientific community due to its far-reaching environmental and socioeconomic impacts. On one hand, more precipitation provides much-needed relief from drought conditions in this region, potentially shifting its climate from a warm-dry to a warm-wet regime, both in summer and annually[9–11]. On the other hand, the increase poses new challenges for water resource risk management. The region has experienced a rise in extreme precipitation events[10–12], leading to numerous floods and other rainfall-

related hazards in recent years, resulting in the loss of life and property[13].

Many studies have attempted to understand the root cause of this wetting trend. It has been interpreted as a superposition of large-scale circulation changes (dynamic)[14–18] and long-term atmospheric moistening (thermodynamic)[8,19]. However, the question of whether this trend is driven by internal variability or external forcings remains unexplored, complicated by the intricate link between these two factors, making a clean separation challenging. Bridging this knowledge gap is crucial for comprehending the physical mechanisms underlying the historical trend and for providing more robust projections of future precipitation changes in the region[10,20–22]. In this study, we address this problem by employing a hierarchy of numerical simulations conducted with Geophysical Fluid Dynamics Laboratory (GFDL) climate models. Our findings attribute the observed trend to atmospheric internal variations as the primary driver.

[1]Cooperative Programs for the Advancement of Earth System Science, University Corporation for Atmospheric Research, Boulder, CO, USA. [2]NOAA/Geophysical Fluid Dynamics Laboratory, Princeton, NJ, USA. [3]Schiller Institute for Integrated Science and Society and Department of Earth and Environmental Sciences, Boston College, Boston, MA, USA. [4]School of Earth and Atmospheric Sciences, Georgia Institute of Technology, Atlanta, GA, USA. [5]Department of Environmental Science and Engineering, California Institute of Technology, Pasadena, CA, USA. ✉e-mail: Wenhao.Dong@noaa.gov

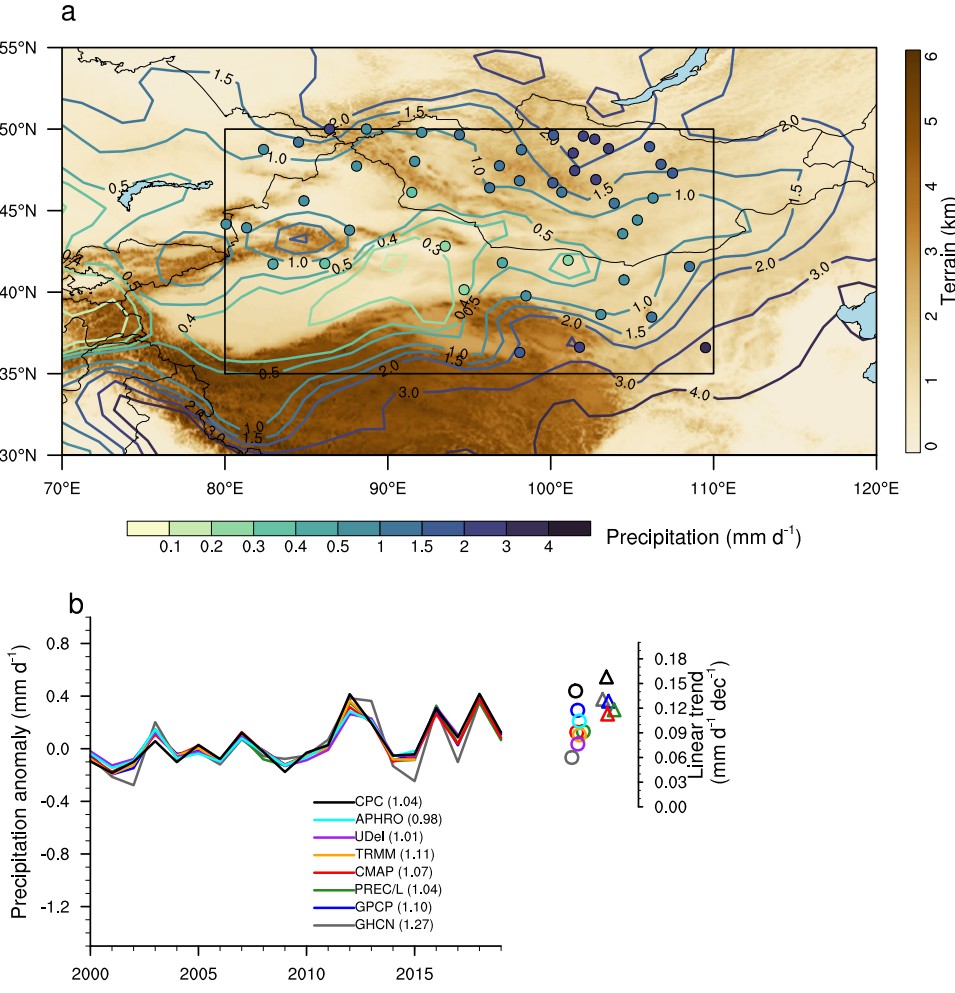

**Fig. 1 | Observed summer precipitation features. a** Topographic features overlaid with long-term mean precipitation (mm d$^{-1}$) based on the NOAA/Climate Prediction Center (CPC) dataset (contours) and the Global Historical Climatology Network (GHCN) station records (dots) over the inner Eurasian continent. The subregion of Taklamakan and Gobi Desert (TGD) is delineated by the black rectangle. **b** Time series of TGD-average summer precipitation anomalies (units: mm d$^{-1}$) based on multiple datasets (see Methods) during 2000–2019. The anomalies are calculated by subtracting the mean value during the overlapping period of 2000–2014 (indicated by the number in the parentheses). Linear trends of TGD-average summer precipitation (units: mm d$^{-1}$ dec$^{-1}$) during 2000–2014 (circles) and 2000–2019 (triangles) are included. The trends are slightly shuffled horizontally for better visualization. Source data are provided as a Source Data file.

## Results

### Observed and simulated precipitation trends

We focus on the summer (June-July-August) precipitation during 2000–2019 when the observed wetting trend over TGD (delineated by the black rectangle in Fig. 1a) is prominent. Figure 1a shows the spatial distribution of the 20-year mean summer precipitation, characterized by a sharp contrast between the outer mountains (~2 mm d$^{-1}$) and inner basins (~0.5 mm d$^{-1}$). The associated interannual variations are highly consistent among different gridded datasets (Fig. 1b), with correlation coefficients higher than 0.94 ($p < 0.01$) during their overlapping years. The mean precipitation based on station records is consistently higher than the gridded datasets likely due to the stations being located in or close to the relatively wet mountainous regions (Fig. 1a). Nonetheless, the gridded datasets and station records are strongly correlated at the interannual time scale. All gridded datasets exhibit significant wetting trends during 2000–2019, ranging from 0.11 to 0.16 mm d$^{-1}$ dec$^{-1}$, or about 10% dec$^{-1}$ of the long-term mean. The station records yield a comparable wetting trend (0.13 mm d$^{-1}$ dec$^{-1}$) during the same period (Fig. 1b and Supplementary Fig. S2a).

In comparison to observations, all three members of the CMIP experiment are able to capture the spatial distribution of the TGD summer precipitation, albeit with slight overestimation over the mountainous regions (Supplementary Fig. S1). These wet biases are common to CMIP5[23] and CMIP6 models[22], and might be related to the representation of topographic features over this region. Nevertheless, in terms of long-term mean and spatial distribution, results from GFDL model simulations outperform the majority of the CMIP6 models (Supplementary Fig. S3), adding confidence in their suitability for comprehending precipitation variability in the TGD region. The ensemble members, however, differ considerably in terms of their linear trends (Supplementary Fig. S2). While the ensemble-mean reports a wetting trend when averaged over TGD (0.18 mm d$^{-1}$ dec$^{-1}$), it is not statistically significant ($p > 0.1$), and the values for the individual ensemble members range from 0.06 to 0.27 mm d$^{-1}$ dec$^{-1}$ (Fig. 2a). The simulated precipitation in the CMIP experiment can be viewed as a combination of internal variability and forced response. External forcing typically operates over longer time scales due to gradual or persistent changes over time. Internal variability tends to dominate over relatively shorter time scales, driven by internal processes and feedback mechanisms within the climate system that can vary rapidly and chaotically. Notably, internal variability-induced changes can contribute significantly to climate changes, and even surpass externally-forced changes on local to continental scales, particularly in mid- and high-latitude regions[24–26]. This influence may persist over several decades. The large spread across the three CMIP ensemble members suggests the possibility of

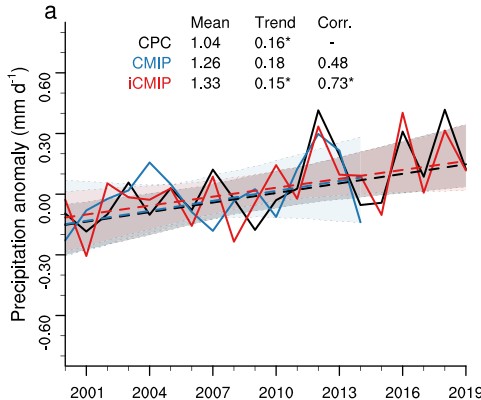
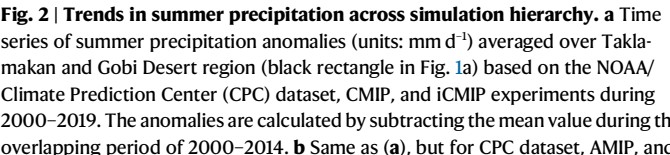
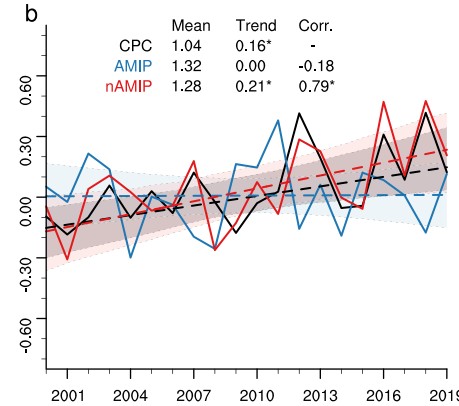

**Fig. 2 | Trends in summer precipitation across simulation hierarchy. a** Time series of summer precipitation anomalies (units: mm d⁻¹) averaged over Taklamakan and Gobi Desert region (black rectangle in Fig. 1a) based on the NOAA/Climate Prediction Center (CPC) dataset, CMIP, and iCMIP experiments during 2000–2019. The anomalies are calculated by subtracting the mean value during the overlapping period of 2000–2014. **b** Same as (**a**), but for CPC dataset, AMIP, and nAMIP experiments. Note, the CM4 experiment ends in the year 2014. The best linear fit and prediction errors are represented by the dashed line and shading, respectively. The long-term mean (units: mm d⁻¹), linear trend (units: mm d⁻¹ dec⁻¹), and the correlation coefficient (units: dimensionless) with the CPC dataset are listed. Linear trends with an asterisk denote statistical significance at the 95% confidence level. Source data are provided as a Source Data file.

internal variability playing an important role in shaping the observed wetting trend.

### Relative role of internal variability and external forcings

To assess the relative influences of internal variability and external forcings on the observed wetting trend, we construct the density distribution of the 20-year linear trends of TGD summer precipitation by resampling the piControl experiment, where the simulated precipitation is solely a response to internal variability (See Methods). As shown in Fig. 3a, the density distribution approaches Gaussian based on the Kolmogorov–Smirnov two-sample test, without a discernible mean trend. The observationally-based estimates fall within the 95% confidence interval of the simulated trends, suggesting that internal variability is a plausible explanation for the observed wetting trend. This inference is further supported by the variability analogue analysis (See Methods). The resulting analogue ensemble mean based on 10-year segments matches both the spatial pattern and the temporal evolution of the observed trend reasonably well (Fig. 3b, c). Notably, the pattern correlation for the simulated trend across TGD is 0.74 ($p < 0.01$). Moreover, the average wetting trend is 0.13 mm d⁻¹ dec⁻¹, and the correlation coefficient of interannual variability is 0.95 ($p < 0.01$). Similar results are obtained with 5-year segments (Fig. 3a and Supplementary Fig. S3). These results strongly indicate that the observed wetting trend could be reasonably attributed to internal variability.

The iCMIP experiment provides a more explicit assessment. It maintains the same external forcing as the CMIP experiment but is distinguished by a more realistic representation of internal variability (See Method). In contrast to CMIP, iCMIP demonstrates skills in reproducing both the observed wetting trend (0.15 mm d⁻¹ dec⁻¹; $p < 0.05$) and the interannual variation ($R = 0.73$; $p < 0.05$) of TGD summer precipitation (Fig. 2a and Supplementary Fig. S2). This finding demonstrates that when appropriately initialized with specific atmospheric and oceanic states, a dynamic model like the one employed in this study has the capability to predict the time evolution of the weather/climate system (i.e., internal variability) in the TGD region within a defined limit, beyond which memory loss occurs due to a combination of error amplification and model limitations. The fact that the model's ability to replicate the observed wetting trend relies on initialization, rather than external forcings, strongly implies that the trend is a result of internal variability. Furthermore, we note that no predictive skill survives an increase in lead time from zero to one month, indicating that memory loss occurs within one month.

### Atmospheric internal variability versus underlying SST/Sea-ice condition

The comparison between CMIP and iCMIP experiments highlights the role of internal variability in reproducing the observed wetting trend. Nonetheless, the core driver behind this trend—whether it stems primarily from atmospheric internal variability or the underlying SST/Sea-ice condition—remains ambiguous. This question bears crucial implications for subseasonal-to-seasonal prediction, as the atmosphere and ocean exhibit markedly distinct memory characteristics. We use the AMIP experiment, in which the oceanic conditions can be thought of as perfect predictions, to isolate the role of atmospheric internal variability (See Method). Note that the direct response to external forcings is confined to the atmosphere and land in AMIP, and may be weaker than the full response in CMIP[27]. Despite showing reasonably good climatology (Supplementary Fig. S1), the AMIP experiment fails to simulate the observed wetting trend and instead shows no discernible trend (Fig. 2b). The interannual variations also differ substantially from the observations ($R = 0.11$, $p > 0.1$).

In contrast, the nAMIP experiment, where winds are nudged to observation (see Methods), shows considerable skills in reproducing both the observed wetting trend (0.21 mm d⁻¹ dec⁻¹; $p < 0.05$) and interannual variations ($R = 0.79$; $p < 0.05$) (Fig. 2b). These results suggest that the primary driver of the wetting trend is atmospheric internal variability, while the role of oceanic conditions, if any, is limited. This conclusion is consistent with previous analyses conducted over North America, where it was found that internal variability in the large-scale atmospheric circulation played a substantial role in explaining variations in precipitation[28–30]. Collectively, these findings provide compelling support for attributing the observed wetting trend to internal variability in the atmosphere.

### Potential atmospheric variability source

Intrinsic atmospheric variability can originate from nonlinear physical and/or dynamical processes. While these fluctuations may be limited to a few weeks, the atmospheric circulation can exhibit longer time-scale variations[24,31]. As one of the leading recurring modes of internal atmospheric variability over the Northern Hemisphere, the North Atlantic Oscillation (NAO) and its interaction with the Eurasian wave train[32,33] are worth further investigation. The summer NAO index has been predominantly in the negative phase since the mid-2000s[34,35], which seems to be consistent with the observed wetting trend over TGD. The phase change in the NAO has been associated with the shift of the North Atlantic summer storm tracks, with a negative phase

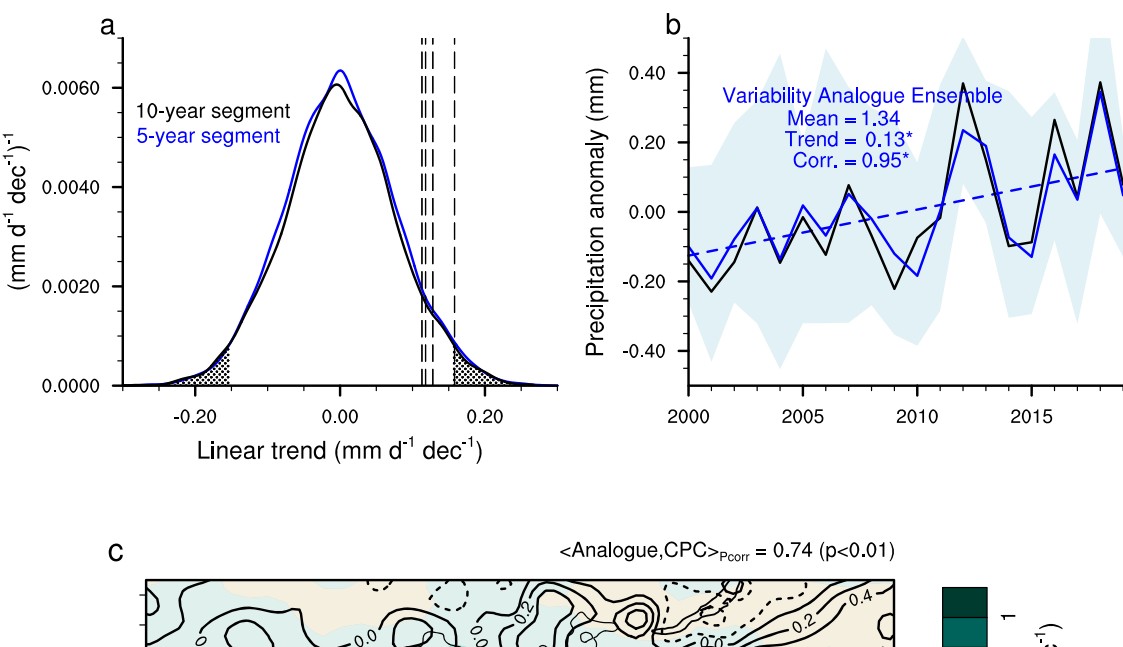

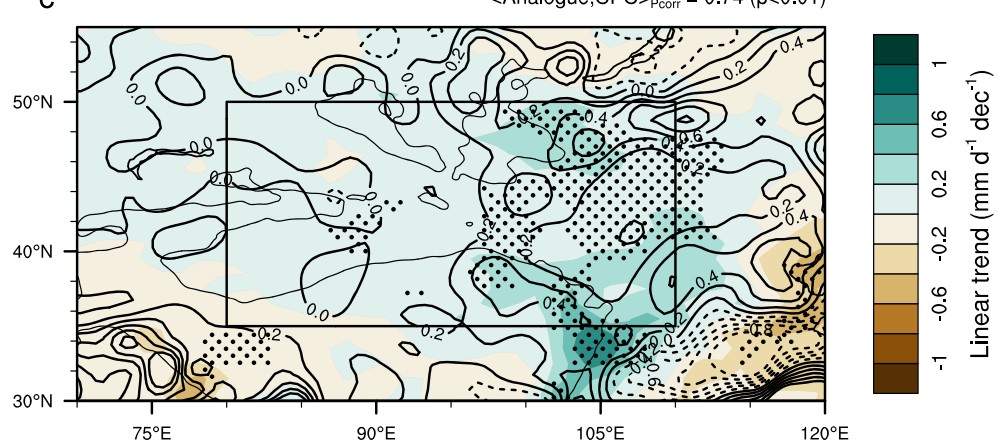

**Fig. 3 | Characteristics of summer precipitation based on the piControl experiment. a** Kernel probability density estimates of the linear trends of 20-year segments based on the resampled piControl simulation. The dashed vertical lines indicate the observed trends and dotted regions represent trends that fall outside the 95% confidence intervals. **b** Time series of the summer precipitation anomalies (units: mm d⁻¹) averaged over Taklamakan and Gobi Desert (TGD; black rectangle in **c**) based on the variability analogue ensembles using 10-year segments (blue) and the NOAA/Climate Prediction Center (CPC) dataset (black). The anomalies are calculated by subtracting the mean value during the entire 20 years. The light blue shading denotes the spread among the respective ensembles. The mean, linear trend, and correlation coefficient with the CPC dataset are shown for variability analogue ensembles. **c** Linear trends of summer precipitation (units: mm d⁻¹ dec⁻¹) based on the variability analogue ensembles using 10-year segments (shading) overlaid with the trends based on CPC dataset (contours). Dashed contours represent negative trends while solid contours indicate positive trends. The gray isoline in (**c**) is the 2000-m contour of surface elevation. Stippling in (**c**) indicates where the linear trends are statistically significant at the 95% confidence level. Source data are provided as a Source Data file.

linked to a southward displaced and zonally elongated storm track, leading to more extratropical storms traveling across northwestern Europe while less frequent storm activity in the Mediterranean region[36–39]. Such a linkage is evident in the correlation coefficient map between the summer NAO index and storm activity (Fig. 4b), which shows a dipole structure over the North Atlantic, characterized by negative anomalies south of Greenland and positive anomalies along the western coast of Europe. Consistent with many previous studies[36,39], this dipole pattern aligns closely with the first leading mode of the empirical orthogonal function (EOF1) analysis of storm activity (Supplementary Fig. S5). The corresponding time series (PC1) exhibits a significant negative correlation ($R = -0.77$; $p < 0.01$) with the summer NAO index (Fig. 4a).

Similar to North Atlantic, we also observe a distinct dipole pattern in the correlation map between summer NAO and storm activity around TGD–another significant center of storm activity (Fig. 4c). When applying the same EOF analysis, the second leading mode (EOF2) reveals a similar dipole structure (Supplementary Fig. S6), and the corresponding time series (PC2) is negatively correlated with the summer NAO index ($R = -0.67$; $p < 0.01$; Fig. 4b). This result implies

the observed wetting trend over TGD may be associated with the southward shift of the storm activities, influenced by the summer NAO. We further analyze the co-evolution of changes in storm activity, precipitation, geopotential height, and horizontal winds during 2000–2007 and 2008–2019, accounting for the significant change in the summer NAO around 2007. Mean precipitation during the latter period is systematically larger than that during the former over TGD, and the spatial pattern of this increase closely resembles the observed wetting trend (Fig. 4d). This enhanced precipitation is consistent with changes in storm activities, which shows a substantial increase over TGD (Fig. 4e). The southward shift of storm track closely relates to upper-level anticyclonic anomalies and positive geopotential height anomalies upstream over central Europe (Fig. 4f), which could trigger eastward propagating Rossby waves on the intraseasonal timescale[37]. In summary, the mechanism underlying the linkage between the summer NAO and the precipitation over TGD mirrors what has been observed in Western Europe. In both cases, the primary controlling factor for variations in precipitation is the synoptic-scale storm activity, which is significantly influenced by the summer NAO[36,39].

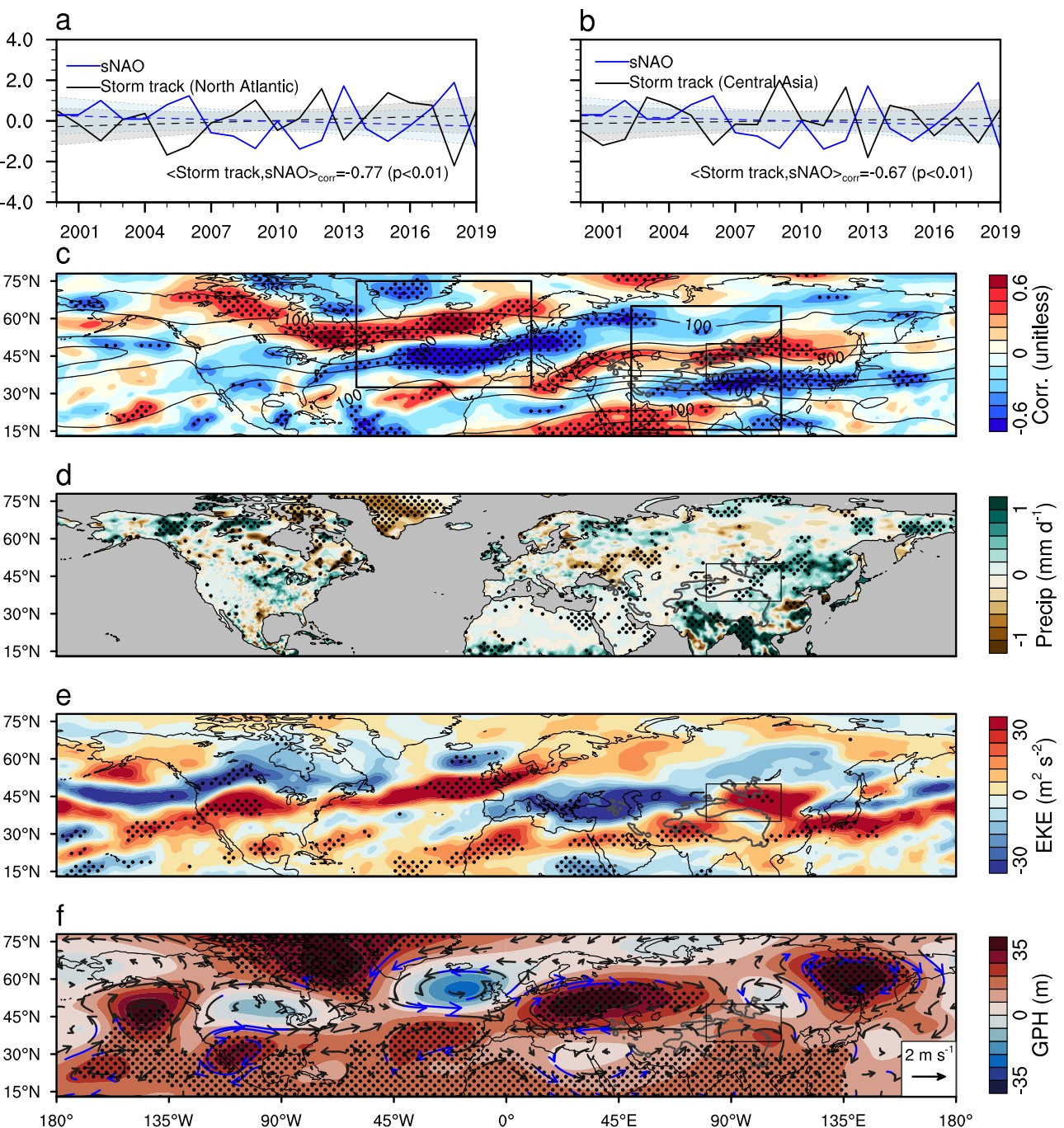

**Fig. 4 | Linkage between the summer North Atlantic Oscillation (NAO), storm activity, and precipitation.** Time series of summer NAO index and time series of the first leading mode of storm activity over North Atlantic (**a**) and time series of the second leading mode of storm activity over Central Asia (**b**). The best linear fit and prediction errors are represented by the dashed line and shading, respectively. **c** Correlation coefficient (units: dimensionless) between summer NAO index and the summer storm activity over the Northern Hemisphere during 2000–2019. Thick black rectangles represent North Atlantic and Central Asia used for the EOF analysis, respectively. **d**–**f** Difference in summer precipitation (units: mm d⁻¹), storm tracks (units: m² s⁻²), and 200 hPa geopotential height (units: m) and horizontal winds (units: m s⁻¹) between 2008–2019 and 2000–2007 (the former minus the latter). In (**c**–**f**), the Taklamakan and Gobi Desert (TGD) region is delineated by the black rectangle. The gray isoline represents the 2000-m contour of surface elevation. Stippling/blue vector in (**c**–**f**) indicates where the differences are statistically significant at the 95% confidence level. The result in (**d**) is based on the NOAA/Climate Prediction Center (CPC) dataset while results in the remaining panels are based on the Fifth Generation of the European Centre for Medium-Range Weather Forecasts (ECMWF) Reanalysis (ERA5). Source data are provided as a Source Data file.

## Verification on a synoptic-scale

To validate that the simulated wetting trends in iCMIP and nAMIP experiments are physically correct, it is critical to assess if they can adequately simulate the associated synoptic scale features. We compare the distinct precipitation patterns and the associated synoptic features in both experiments with observations based on a clustering method (See Methods). Remarkably, both experiments successfully capture the observed major precipitation patterns (Fig. 5). Specifically, the first cluster features negative precipitation anomalies over large parts of TGD, while the second and third clusters exhibit significant

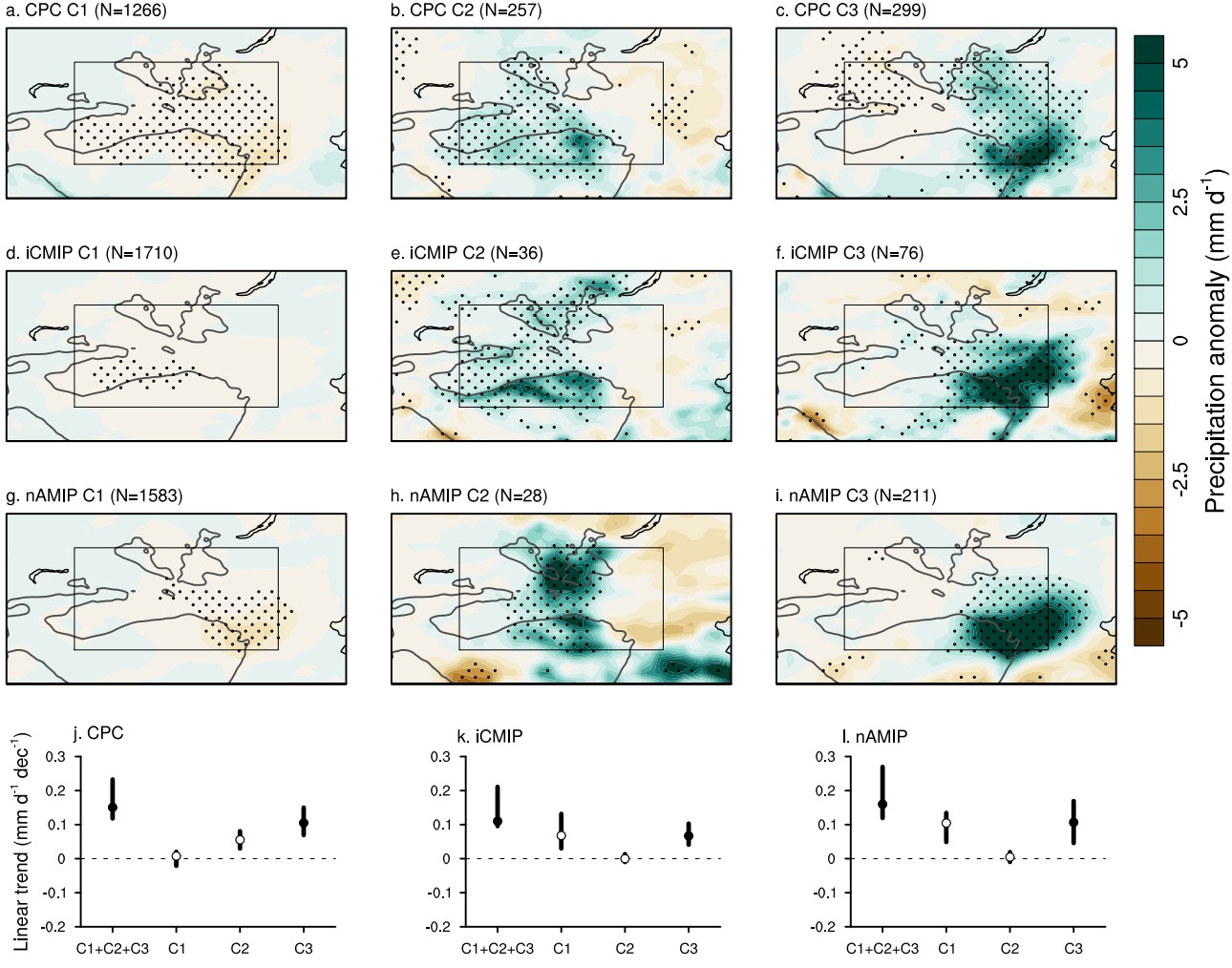

**Fig. 5 | Hierarchical clustering analysis of summer precipitation.**
**a**–**c** Precipitation anomalies (units: mm d$^{-1}$) for the three identified clusters based on the NOAA/Climate Prediction Center (CPC) dataset. **d**–**f** Same as (**a**–**c**), but based on the iCMIP experiment. **g**–**i** same as (**a**–**c**), but based on the nAMIP experiment. The Taklamakan and Gobi Desert (TGD) region is delineated by the black rectangle. The gray isoline represents the 2000-m contour of surface elevation. Linear trends of summer precipitation (units: mm d$^{-1}$ dec$^{-1}$) in each cluster averaged over TGD based on (**j**) the CPC dataset, (**k**) iCMIP, and (**l**) nAMIP experiments. $N$ in (**a**–**i**) is the number of samples in each cluster. Stippling in (**a**–**i**) indicates regions where the anomalies are statistically significant at the 95% confidence level. The gray isoline in (**a**–**i**) is the 2000-m contour of surface elevation. Solid dots in (**j**–**l**) denote the linear trends are statistically significant at the 95% confidence level. The thick vertical lines in (**j**–**l**) are the 95% confidence intervals for each cluster. Source data are provided as a Source Data file.

positive anomalies over the central and southeastern parts of TGD, respectively. Pattern correlations between both experiments and observations are statistically significant ($p < 0.01$) for all three clusters. Furthermore, in line with observational data, the probability density distributions demonstrate that the latter two clusters are distinguished by their association with intense precipitation occurrences (Supplementary Fig. S7). The corresponding geopotential height anomalies for these events reveal clear wave train patterns traversing the Eurasian continent (Supplementary Fig. S8), and the associated moisture flux patterns favor the occurrence of these large precipitation events (Supplementary Fig. S9). Concurrently, the storm activities align well with these large precipitation events (Supplementary Fig. S10). The nAMIP and iCMIP simulations capture the general synoptic features, but notable biases are evident when compared to observations. Based on the geopotential height anomaly, the slightly poleward-shifted wave pattern in the second cluster is distorted in the nAMIP simulation, while the zonal wave pattern in the third cluster is misrepresented in the iCMIP simulation. Similar discrepancies are observed in the low-level moisture fluxes. These biases may be partly attributed to the smaller sample size in the model simulation, warranting further investigation. More importantly, when we compare the relative

contributions of the three clusters to the total precipitation trend, it becomes evident that both the observed and simulated wetting trend primarily arises from the third cluster, characterized by strong precipitation events occurring over the eastern TGD (Fig. 5).

## Discussion

Using a hierarchy of numerical simulations and a variety of statistical analysis tools, we offer unequivocal evidence that the recent wetting trend observed over TGD is substantially influenced by internal variability, rather than the prescribed boundary conditions (e.g., sea surface temperatures/sea ice and radiative forcings). In particular, the recent shift of the summer NAO towards its negative phase has led to a southward displacement of storm activities over TGD. These changes, in turn, facilitate the passage of a greater number of extratropical storms into TGD, thereby resulting in an increased precipitation trend. This mechanism bears a notable resemblance to the patterns observed over Western Europe, where North Atlantic storm activities, governed by the summer NAO, exert substantial control over the regional precipitation patterns. Our clustering analyses further confirm that these connections predominantly manifest at the synoptic scale, with larger contributions to the wetting trend originating from large precipitation

events. The dominant role of internal variability in driving the historical wetting trend suggests that it may not be sustainable and is likely to reverse in the future, thus the prospect of a new, wetter climate regime[10,20] in this historically arid region may be illusory. To comprehensively grasp TGD's future precipitation trends, it is imperative to account for internal variability alongside anthropogenic forcing. Furthermore, the hierarchical model simulation framework employed in this study demonstrates its robust utility as a powerful tool for conducting attribution analyses related to both internal variability and external forcing. This framework can be effectively extended to other regions, with the added advantage of requiring less computational resources compared to conventional large ensemble simulations.

## Methods

### Observational datasets

Multiple widely used gridded monthly precipitation datasets, including NOAA/Climate Prediction Center (CPC) precipitation dataset[40], U of Delaware (UDel) dataset[41], Tropical Rainfall Measuring Mission (TRMM) 3B43[42], CPC Merged Analysis of Precipitation (CMAP) dataset[43], NOAA's Precipitation Reconstruction over Land (PREC/L) dataset[44], Global Precipitation Climatology Project (GPCP) dataset[45], and Asian Precipitation-Highly-Resolved Observational Data Integration Towards Evaluation (APHRODITE; shorted for APHRO) dataset[46] are compared in this study to examine the wetting trend, among which the CPC precipitation dataset is adopted as the main one since the daily coverage makes it uniquely suitable for the subsequent clustering analysis (to be detailed). Precipitation measurements from 47 observation stations located in Northwest China, Kazakhstan, and Mongolia, retrieved from the Global Historical Climatology Network monthly (GHCNm) precipitation dataset, are utilized as a reference (see Fig. 1a for their geographical locations). The synoptic-scale analysis is based on daily specific humidity, geopotential height, and large-scale winds from the ERA5 reanalysis[47]. The principal component (PC)-based indices of the North Atlantic Oscillation (NAO)[48] are used in this study. This index is derived through empirical orthogonal function (EOF) analysis, which decomposes sea level pressure anomalies over the Atlantic sector (20°–80°N, 90°W–40°E) into a series of uncorrelated and orthogonal patterns. The primary EOF pattern, characterized by a meridional dipole between Iceland/Greenland and the Azores, represents the NAO pattern. The associated time series of the principal component (PC), obtained by projecting the EOF onto the anomaly field, represents the NAO index.

### Model simulations

A hierarchy of global climate model simulations are performed with the different configurations of the NOAA/GFDL atmospheric model AM4[49,50]. AM4, with a horizontal resolution of ~100 km and 33 vertical levels, is the atmospheric component of the GFDL coupled physical climate model CM4[51], climate prediction model SPEAR[52], and Earth system model ESM4.1[53]. CM4 and ESM4.1 participate in the Coupled Model Intercomparison Project Phase 6 (CMIP6), and SPEAR contributes to the North American Multi-Model Ensemble (NMME)[54]. Note, SPEAR can run with different atmospheric horizontal resolutions ranging from 1° to 0.25°. We have chosen to use the medium resolution configuration (i.e., SPEAR-MED) due to the availability of large ensembles. A comparison with SPEAR-LO reveals minor differences in simulated precipitation and large-scale fields, suggesting that variations in horizontal resolution have relatively small impacts. This series of GFDL models are generally more skillful than the previous generations of GFDL models and other CMIP6 models in simulating both the mean climate state[55] and various modes of climate variability and extreme weather events such as monsoon low-pressure systems[56], mesoscale convective systems[57–59], atmospheric rivers[58,60,61], and tropical cyclones[58,62,63]. To further provide a broader context for the model results, we have included a comparison of precipitation

simulation over the TGD region from 40 CMIP6 models (see model details in Supplementary Table S1). As shown in Supplementary Fig. S3, when compared with the CPC precipitation dataset, GFDL models used in the study exhibit smaller precipitation biases and higher pattern correlations than the majority of CMIP6 models.

Four types of simulations are used to separate the roles of internal variability and external forcings. A schematic outlining these simulations, along with the rationale behind each simulation, is provided in Supplementary Fig. S11. The first type involves the three-member fully coupled CM4 ensemble (referred to as CMIP), driven by historical anthropogenic and natural forcings such as greenhouse gases, aerosols, ozone, land use, solar irradiance, and volcanic aerosols. The second type, initialized CMIP simulations (iCMIP), is part of the seasonal prediction experiments conducted with SPEAR[64]. For iCMIP, atmospheric initial conditions are derived by nudging the model to reanalysis data for three-dimensional temperature, winds, and specific humidity, while the oceanic counterparts are generated with an ensemble-based ocean data assimilation system. We use 15 members (realizations), with each simulation lasts 12 months. The lead time is set to 0, meaning that the mean precipitation for a specific month (e.g., July 2005) is computed by averaging the simulations initialized on the first day of that month (e.g., July 1, 2005). The three-member CMIP and 15-member iCMIP simulations are created by slightly perturbing the initial conditions of each respective model.

Following the Atmosphere Model Intercomparison Project (AMIP) protocol, the third type (AMIP) consists of atmospheric-only simulations forced with historical sea surface temperatures (SSTs), sea ice, and other external forcings, identical to those used in CMIP. However, in the AMIP setup, the forcings can only affect the atmosphere and land as the oceanic conditions are prescribed. In the fourth type, the nudged AMIP simulation (nAMIP), large-scale winds are nudged to reanalysis data, rather than allowing the atmospheric model to generate its own winds. This nudging setup is similar to that used by an earlier study[65], but only large-scale winds are nudged in nAMIP. It's important to note that specific humidity, clouds, and precipitation are computed interactively in both AMIP and nAMIP simulations, subject to the same dynamic and physical processes as in the AMIP setup. Other variables, such as temperature and humidity, are not nudged in this study. This decision is made to achieve a balance between model dynamics/physics and observational constraint, as incorporating additional nudging variables can sometimes lead to significant inconsistencies in the mean climate. AMIP and nAMIP each have one realization. All simulations are conducted with identical external forcings and are analyzed at the same 100-km horizontal resolution.

The CMIP simulations cover the period of 2000–2014, while AMIP, iCMIP, and nAMIP span 2000–2019. These simulations are analyzed and compared with observations for their entire durations. However, it is worth mentioning that shortening the analysis to 2000–2014 for AMIP, iCMIP, and nAMIP yields similar results. Additionally, we employ the CM4-based preindustrial unforced control (piControl) simulation with a length of 650 years to conduct the unforced trend analysis and variability analogue analysis (details provided below).

### Clustering analysis

A hierarchical clustering algorithm is employed to all summer precipitating days over TGD to identify predominant spatial patterns[66]. Each precipitating day, defined as falling within the 99th percentile of the overall domain-average daily precipitation distribution, is initially assigned to one cluster. The similarity between each single-member cluster is calculated using Ward's minimum variance method[67], which measures the Euclidean distance between each pair of clusters. The clustering algorithm then merges the pair of clusters with the highest degree of similarity and repeats this process until all precipitation days become members of one single cluster. To determine the optimal

number of clusters, we apply the average silhouette approach. As a result, we identify a total of three clusters for both the observation and the model simulations. The results are consistent if we extend the summer season to warm season (May–September) and over a longer period (1979–2019), attesting to the robustness of this method in identifying the prevailing spatial patterns of precipitation. The anomalies in moisture transport and divergence, and geopotential height associated with each cluster are constructed by compositing the respective daily anomalies.

### Unforced trend analysis and variability analogue analysis

To construct the density distribution of linear trends based on the unforced control simulation, we utilize a resampling approach by first combining two randomly selected 10-year segments and then computing the trend of the combined 20-year period using the Theil-Sen estimator. The control simulation is linearly detrended to remove any residual model drift before resampling. The selection of 10-year segments (as opposed to a 20-year segment) is justified in the sense that any linear trends due to decadal variability can be largely preserved while also accommodating the relatively short piControl simulation. Note that four 5-year segments yield similar results. A total of 10,000 samples are derived from repeating the above procedure with replacement. This resampling helps increase the sample size and provides a more reliable estimation.

Variability analogue analysis, introduced by a previous work[68], is employed in this study. This method uses segments from unforced control simulations to match observations. Based on the resulting analogue ensembles, one can estimate the contribution of natural variability to the observed variability. This method has been successfully used in previous studies examining observed temperature trends[69] and ENSO variability[68]. To identify analogous segments from the resampled 20-year data, we select those that exhibit a resemblance to the evolution of precipitation over TGD during 2000–2019. Applying a correlation coefficient ($R$) threshold of 0.6 yields 33 out of 10,000 segments based on two 10-year segments, and 48 out of 10,000 segments based on four 5-year segments. Note that 0.6 is chosen to achieve a balance between the strength of correlation and the number of the variability analogue ensemble members[68].

### Storm activity analysis

Storm activity in this study is measured using the upper-level (200 hPa) transient eddy kinetic energy (EKE), following the methodology established earlier[70]. EKE is calculated based on the daily horizontal winds with deviations from their time mean values. To ensure the robustness of our findings, we have cross-validated our results with alternative storm activity proxies, such as bandpass-filtered geopotential height anomalies, which have yielded consistent results. An empirical orthogonal function (EOF) analysis is applied to the upper-level EKE over two distinct regions characterized by the highest levels of storm activities during 2000–2019: The North Atlantic and Central Asia (see black rectangles in Fig. 4c for their respective locations). This analysis enables us to demonstrate their major spatial pattern and interannual evolution of the storm activities in these regions (Supplementary Figs. S5, S6).

Over the North Atlantic region, the spatial distribution of the first leading mode (EOF1) reveals a dipole pattern characterized by negative anomalies situated to the south of Greenland and positive anomalies adjacent to the western coast of Europe (Supplementary Fig. S5). Similarly, in the Central Asia region, the spatial distribution of the second leading mode (EOF2) exhibits dipole structures, with positive anomalies located to the south of the TGD area and negative anomalies to the north (Supplementary Fig. S6). These two dominant patterns are distinctly separable from other modes using the North test method[71]. Compared to the mean storm activities, the positive values of PC1 over the North Atlantic and PC2 over

Central Asia are indicative of a southward shift in storm activities. Notably, the time series data of these principal components exhibit significant ($p < 0.01$) correlations with the summer NAO index (Fig. 4).

### Statistics analysis

Trends are calculated using the robust Theil–Sen estimator[72,73], in which the linear trend represents the median slope between all paired values. The Theil–Sen estimator is designed to reduce the influence of outliers and endpoints in linear trend analysis. Confidence intervals of the median slopes are calculated[73]. Correlation coefficients are calculated using the Pearson correlation with long-term trends removed. Kolmogorov–Smirnov two-sample test is used to determine if two samples are from the same distribution.

### Reporting summary

Further information on research design is available in the Nature Portfolio Reporting Summary linked to this article.

## Data availability

The monthly Tropical Rainfall Measuring Mission (TRMM) precipitation data is accessible at https://disc.gsfc.nasa.gov/datasets/TRMM_3B43_7/summary. The station records of the monthly precipitation dataset are available for acquisition at https://www.ncei.noaa.gov/data/ghcnm/v4beta/. The other precipitation datasets utilized in this study, including the CPC Global Unified Gauge-Based Analysis of Daily Precipitation dataset, the monthly GPCP precipitation dataset, the monthly UDel precipitation dataset, and the monthly CPC Merged Analysis of Precipitation (CMAP) precipitation dataset, can be found and downloaded at https://psl.noaa.gov/data/gridded/index.html. The PC-based summer NAO index is obtained from https://climatedataguide.ucar.edu/sites/default/files/2023-04/nao_pc_monthly.txt. For the ERA5 reanalysis datasets, they can be accessed from https://apps.ecmwf.int/data-catalogues/era5/?class=ea. The GFDL AM4 model source code can be obtained from https://data1.gfdl.noaa.gov/nomads/forms/am4.0/. Model outputs from AMIP, CMIP, and piControl experiments can be downloaded from the CMIP6 data portal (https://aims2.llnl.gov/search/cmip6/). Model outputs from iCMIP experiment can be accessed publicly from the GFDL SPEAR Large Ensembles website (https://noaa-gfdl-spear-large-ensembles-pds.s3.amazonaws.com/index.html#SPEAR/GFDL-LARGE-ENSEMBLES/CMIP/NOAA-GFDL/GFDL-SPEAR-MED/historical/). Model outputs from the nAMIP experiment have been deposited in the Zenodo database under accession code (https://doi.org/10.5281/zenodo.11110869). Source data are provided with this paper.

## Code availability

The NCAR Command Language (NCL v6.6.2) is used for plotting. All custom codes are direct implementations of standard methods and techniques, described in detail in Methods.

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

## Acknowledgements

The authors would like to thank Ming Zhao and Akshaya Nikumbh for their useful discussion and comments on earlier versions of this paper. This research from the Geophysical Fluid Dynamics Laboratory is supported by NOAA's Science Collaboration Program and administered by UCAR's Cooperative Programs for the Advancement of Earth System Science (CPAESS) under awards NA16NWS4620043 (W.D.) and NA18NWS4620043B (W.D.). This work is also support by the U.S. National Science Foundation (NSF) through Grant AGS-2032532 (Y.D.) and by the U.S. National Oceanic and Atmospheric Administration (NOAA) through Grant NA22OAR4310606 (Y.D.) and Grant NA20OAR4310380 (Y.D.).

## Author contributions

W.D. and Y.M. conceived and designed the study. W.D. performed the analyses, with contributions from Y.M. and Y.D. in interpreting the results. Z.S. conducted the nudged simulation. W.D. and Y.M. wrote the paper, and Y.D. and Z.S. contributed to discussions and improving the manuscript.

## Competing interests

The authors declare no competing interests.
