## [Peer Review File · Nature Communications]

Recent Wetting Trend over Taklamakan and Gobi Desert
Dominated by Internal VariabilityREVIEWER COMMENTS

Reviewer #1 (Remarks to the Author):

- (1) The main contribution in this work is to assess the relative role of internal variability and external forcings, but the quantitative results are not clear.
- (2) In Methods, the authors said that GCM simulations are performed with GFDL AM4. But what is the advantage of this model compared to other models. Please let us know the reason for selecting this model.
- (3) In this study, the CM4, ESM4.1 from CMIP6, and other models were used to simulate, but the applicable for these models is not discussed, especially in the TGD.
- (4) What is the three-member fully coupled CM4 ensemble for the first type. Similar problem for other types. For example, iCMIP with 15 members, but the details for these are not discussed.
- (5) For nAMIP, only winds are nudged to reanalysis data? If the difference between AMIP and nAMIP is only from the winds, then the simulation differences between AMIP and nAMIP only reflect the effects of winds on precipitation variations, not all the external forcings.
- (6) Figure 1 should be improved. Where is the definite boundary for the TGD region. Is it only the region within the black rectangle?
- (7) In Figure 1. The authors discussed the different trends in 2000-2014 and 2000-2019. Why? not 2000-2009 vs. 2010-2019, or 2000-2009 vs. 2000-2019, etc. Authors should declare the reasons for the selection of different series. Additionally, the short-term trends are not credible.
- (8) In Figure S1. For i and j subplots, what is the meaning for 0101-0120?
- (9) In Figure S2, we can find that there are dramatically different between the stations and other datasets, especially for the significant trends. It should be more discussed in this work.
- (10) Line 83-89, the further discussion should be added to make the conclusion be reliable and credible.

Reviewer #2 (Remarks to the Author):

This paper uses a series of GFDL model simulations to argue that recent observed trends of increasing precipitation over deserts in interior Asia are due to internal variability, not external forcing such as greenhouse gas emissions. I found the paper analyses compelling and thorough, the writing clear, and the figures helpful. The insight into the trends of this region is relevant for environmental management in arid Asia, and the methodology for distinguishing whether trends are due to internal variability can be usefully applied to other regions in the future. I think this paper will be appropriate for acceptance to Nature Communications following some minor revisions recommended below. Most of these comments are small errors or notes to clarify certain methodologies better. One comment bolded below is more substantial and pertains to the need for further discussion of how reliable (or not) the observed wetting trend is.

1. [44-45] To clarify does “warm-dry” vs “warm-wet” refer to a climate that is like that in the annual mean, or does it mean that the warm season (eg boreal summer) is shifting from dry to wet? If this sentence could be revised slightly to make this clearer that would be helpful.
2. [70] Stations are not shown in Fig. S1a. Do you mean Fig. 1a which does have stations?
3. **[73-75,S2a] In the places where the station records are drying they often look at odds with the gridded dataset according to S2a. The paper seems to be premised on the gridded record being more reliable, since it is focused on pronounced wetting trends. I think some discussion should be added regarding whether the station records or gridded products are more reliable in this region. Do we trust gridded products’ ability to pick up a trend in this region when there aren’t many stations? Relatedly, Fig S2a indicates that the trends tend to be more significant over mountains, and only sporadically significant over the deserts.**
4. [100-101,409-411] Can you please clarify here or in the methods why the ensemble analogue is based on 10 not 20 year segments? That wasn’t obvious to me especially given that the observed record considered is closer to 20 years.
5. [111] I think you mean S2 which has trends. S1 has means.
6. [111-116] I think you should specify somewhere in this sentence that you are referring to what you find in TGD, otherwise it seems like an oddly general statement.
7. [151] Suggest to cut “Especially,”.
8. [199-200] I suggest to soften this statement and perhaps more explicitly acknowledge the biases—it doesn’t appear to me that “all” the synoptic features are simulated reasonably. In particular, iCMIP moisture fluxes appear to be in opposite direction in east part of box for cluster 3. Geopotential height anomalies look somewhat different too.
9. [273-274] This archived manuscript (presumably under review?) seems to have substantial overlaps in the author team and subject matter. For the record, could you please clarify the distinction in subject matter or methodology? I assume they are distinct enough to be published separately I just want to double check.
10. [337-342] I was a bit surprised APHRODITE was not included among the datasets you analyzed since it focuses on high-resolution Asian precipitation. Could you try adding this dataset or explain why it is insufficiently distinct to be needed?
11. [342-344] Please specify where the station-based measurements were retrieved from.
12. [345-348] Please specify a bit more detail about how the NAO index was derived—maybe the observed dataset it was derived from and a citation for the methodology?
13. [353-355] I suggest to rephrase this sentence slightly, since SPEAR is run at a few different horizontal resolutions, but as currently phrased it makes it seem like it is always ~100 km. Are all the simulations you analyze run at ~100 km?
14. [359-361] It would be better if this list focused on extreme weather events that were relevant to the TGD region, or at least noted which ones were relevant to this region.

15. [365-367] Just double checking that the iCMIP simulations with SPEAR are run at 100 km resolution (eg is it SPEAR-LO or SPEAR-MED)?
16. [379-381] What is the timescale of the nudging? Also can you please clarify more specifically in the text what it means that only the large-scale winds are nudged?
17. [396] It's a bit unclear to me what aspects of the clusters is assessed to be similar or not? Is it the spatial pattern of precipitation?
18. [Fig 1a] The contours of precip are hard to see. Can you make them a bit thicker?
19. [Fig 1b] To clarify the time series is just the average in the box, right? No additional masking for dry regions is done?
20. [589] Do you mean "triangles"? I don't see "squares".
21. [593] I suggest to call the start of this caption "Trends in summer precipitation across simulation hierarchy" instead.
22. [596, 610] I think "substrating" should be "subtracting".
23. [Fig 3c] It's not obvious to me why the pattern correlation between the analogue ensemble and the observed data should be high, given that as far as I understand the analogue ensemble is based on area average time series. Am I misunderstanding something and the analogue ensemble includes a spatial component?
24. [614-615] Please clarify in the caption what the dashed vs line contours means.
25. [615-616] You mean that the stippling is where the analogue ensemble linear trends are significant correct?
26. [631] Suggest to call the start of this caption "Cluster analysis of summer precipitation across simulation hierarchy" instead.
27. [637-638] I think this note about the gray isoline referring to topography needs to be added to other figure captions that contour topography too.
28. [SI 20] should say "CPC dataset" not "CPC datasets" since it is one dataset
29. [SI 23] again should say "CPC dataset"
30. [Fig S6] It's weird that precip in mm/day just shows up as xlabel in middle panel. suggest to have at bottom instead or all panels? also what is the y axis unit?
31. [Fig S7] Is the black box TGD? Please note in the caption.

REVIEWER COMMENTS

Reviewer #1 (Remarks to the Author):

Response:

We thank the reviewer for the time invested in careful reading of the manuscript as well as the instructive suggestions. Please see our point-by-point response below:

(1) The main contribution in this work is to assess the relative role of internal variability and external forcings, but the quantitative results are not clear.

Response:

Thank you for bringing up this important point. We understand the reviewer's concern regarding the quantitative results. Isolating forced response from internal variability is always a challenging task since external forcing may change properties of internal variability (e.g., frequency, amplitude, and symmetry). Many previous studies examining the relative roles of internal variability and external forcings rely on large ensembles of coupled simulations (e.g., Dai et al. 2015; Schwarzwald and Lenssen, 2022). In such studies, the ensemble mean typically represents the external forced signal, while the ensemble spread reflects internal variability. However, the specific sources of internal variability (atmospheric versus oceanic) remain poorly understood.

YES/NO denotes whether the observed wetting trend can be simulated.

Figure S11. A schematic of the hierarchical model simulations used in this study. ‘SST/SI state’ indicates that the underlying sea surface temperature (SST) and sea ice (SI) are prescribed, while ‘IC’ denotes the initial conditions for the atmosphere or ocean.

In this study, we approach this problem by separating the internal variability and external forcing gradually using the hierarchical model simulations. We argue that the hierarchical model simulation framework employed here serves as a powerful tool for conducting attribution analyses related to both internal variability (atmospheric versus oceanic) and external forcing. Moreover, this framework can be effectively extended to other regions, offering the added advantage of requiring less computational resources compared to conventional large ensemble simulations. As summarized in the new Fig. S11, our approach unfolds as follows:

(1) We start by examining the simulated precipitation trend over the TGD region in the CMIP experiment. The simulated precipitation in the CMIP experiment results from a combination of internal variability and forced response. When compared with observation, the ensemble mean of CMIP experiments captures the wetting trend (albeit statistically insignificant), but considerable spread exists within the ensemble. Therefore, the CMIP experiment is not sufficient to distinctly attribute the wetting trend to either external forcing or internal variability.

(2) We then switch to the iCMIP experiment, where both the atmospheric and oceanic states are initialized. The comparison between CMIP and iCMIP allows for the isolation of the impact of internal variability, as they share the same external forcing. In contrast to CMIP, iCMIP displays skills in reproducing both the observed wetting trend and the interannual variation of TGD summer precipitation. The fact that the model's ability to replicate the observed wetting trend relies on initialization, rather than external forcings, suggests that the trend arises from internal variability. However, the ambiguity persists regarding the core driver behind this trend—whether it predominantly stems from atmospheric internal variability or underlying SST/sea-ice conditions.

(3) Subsequent examination of the AMIP experiment, where oceanic conditions can be deemed perfect predictions, serves to isolate the role of atmospheric internal variability. However, the AMIP experiment falls short in simulating the observed wetting trend. The interannual variations also diverge substantially from the observations. This suggests that oceanic states play a limited role in the wetting trend, with atmospheric variability likely driving the observed trend.

(4) The inference from AMIP experiment is corroborated in the nAMIP experiment, where horizontal winds in the AMIP simulation are nudged towards observation. Results from the nAMIP experiment demonstrate significant skill in reproducing both the observed wetting trend and interannual variations. These findings underscore the primary role of atmospheric internal variability in driving the wetting trend.

In Step (2), the potential role of internal variability in driving this wetting trend is further supported by analyzing the piControl simulation, where simulated precipitation is solely a response to internal variability. Comparison of trend probability distributions reveals that

observationally-based estimates fall within the 95% confidence interval of simulated trends, suggesting internal variability as a plausible explanation for the observed wetting trend. Besides, an analogue analysis based on the piControl simulation further confirms that the observed wetting trend could be reasonably attributed to internal variability. Building upon these analyses, we delve deeper into the potential drivers of atmospheric variability using clustering analysis. The model's ability to capture different precipitation clusters and their corresponding large-scale circulation patterns reinforces our findings, indicating that the models employed here produce correct results for the right reasons. We have added a new Fig. S11 in the revision to outline these simulations and provide the reasoning behind each one.

Reference:

Dai, A., Fyfe, J. C., Xie, S. P., & Dai, X. (2015). Decadal modulation of global surface temperature by internal climate variability. Nature Climate Change, 5(6), 555-559.

Schwarzwald, K., & Lenssen, N. (2022). The importance of internal climate variability in climate impact projections. Proceedings of the National Academy of Sciences, 119(42), e2208095119.

(2) In Methods, the authors said that GCM simulations are performed with GFDL AM4. But what is the advantage of this model compared to other models. Please let us know the reason for selecting this model.

Response:

The AM4, CM4, and SPEAR are the latest generation of the atmospheric model (Zhao et al., 2018), climate model (Held et al., 2019), and seasonal prediction model (Delworth et al., 2020) developed at GFDL. They are widely recognized as top-tier state-of-the-art global climate models (GCMs), exhibiting superior overall performance compared to other GCMs. While these models have not been evaluated specifically for our study region, they, along with previous generations, have been extensively used in the research community for studying hydroclimate dynamics in similar regions. For instance, Kapnick et al. (2014) used the GFDL CM2.5 model to explore the sensitivity of snowfall to warming in the Karakoram range, demonstrating the model's accurate representation of the region's hydroclimate. GFDL FLOR, another earlier coupled model version, has been employed to investigate the impact of Tien Shan Mountain on precipitation seasonality (Baldwin and Vecchi, 2016), as well as extreme precipitation and landslide occurrences in the Himalayas (Kirschbaum et al., 2020). Furthermore, our earlier study utilized GFDL AM4 to analyze both rainfall and snowfall patterns over the Tibetan Plateau (Dong et al., 2022). These collective studies underscore the GFDL models' capability to simulate precipitation characteristics across diverse and complex terrains, including the High Mountain Asia region and adjacent desert areas.

To provide a broader context for our model results and take your #3 comment into consideration, we have included a comparison of precipitation simulation over the TGD region from 40 CMIP6 models (see model details in Table S1). As shown in Fig. S3, when compared with observation, GFDL models used in the study exhibit smaller precipitation

biases (0.49 mm d⁻¹ based on CMIP6 multi-model mean versus 0.28 mm d⁻¹ based on the average of GFDL models used in our study) and higher pattern correlations than the majority of CMIP6 models. We have incorporated this comparison in the revision (P3L78–83):

“These wet biases are common to CMIP5²⁴ and CMIP6 models²³, and might be related to the representation of topographic features over this region. Nevertheless, in terms of long-term mean and spatial distribution, results from GFDL model simulations outperform the majority of the CMIP6 models (Fig. S3), adding confidence in their suitability for comprehending precipitation variability in the TGD region.”

and (P12L419–424):

“To further provide a broader context for the model results, we have included a comparison of precipitation simulation over the TGD region from 40 CMIP6 models (see model details in Table S1). As shown in Fig. S3, when compared with the CPC precipitation dataset, GFDL models used in the study exhibit smaller precipitation biases and higher pattern correlations than the majority of CMIP6 models.”

Reference:

- Kapnick, S. B., Delworth, T. L., Ashfaq, M., Malyshev, S., & Milly, P. C. (2014). Snowfall less sensitive to warming in Karakoram than in Himalayas due to a unique seasonal cycle. Nature Geoscience, 7(11), 834-840.*
- Baldwin, J., & Vecchi, G. (2016). Influence of the Tian Shan on arid extratropical Asia. Journal of Climate, 29(16), 5741-5762.*
- Kirschbaum, D., Kapnick, S. B., Stanley, T., & Pascale, S. (2020). Changes in extreme precipitation and landslides over High Mountain Asia. Geophysical Research Letters, 47(4), e2019GL085347.*
- Dong, W., & Ming, Y. (2022). Seasonality and variability of snowfall to total precipitation ratio over high mountain Asia simulated by the GFDL high-resolution AM4. Journal of Climate, 35(17), 5573-5589.*

Figure S3. a, Long-term mean of summer precipitation (units: mm d⁻¹) based on CMIP6 multi-model mean. The TGD region is delineated by the black rectangle. The gray isoline represents the 2,000-m contour of surface elevation. b, Boxplot of mean bias of summer precipitation averaged over the TGD region based on 40 CMIP6 models. Results from GFDL models used in this study are denoted by green circles and are slightly shuffled horizontally for better visualization. c, Similar to (b), but for the precipitation pattern correlation calculated over the TGD region. Mean bias and pattern correlation are calculated relative to the CPC precipitation dataset.

(3) In this study, the CM4, ESM4.1 from CMIP6, and other models were used to simulate, but the applicable for these models is not discussed, especially in the TGD.

Response:

Please see our response to your comment #2.

(4) What is the three-member fully coupled CM4 ensemble for the first type. Similar problem for other types. For example, iCMIP with 15 members, but the details for these are not discussed.

Response:

The three-member fully coupled CM4 simulations and the 15-member SPEAR simulations are ensembles generated from the CM4 model and SPEAR-LO model, respectively. These ensembles are created by slightly perturbing the initial conditions of each respective model. We have incorporated this information into the revised text (P12L437–438): “*The three-member CMIP and 15-member iCMIP simulations are created by slightly perturbing the initial conditions of each respective model.*”

(5) For nAMIP, only winds are nudged to reanalysis data? If the difference between AMIP and nAMIP is only from the winds, then the simulation differences between AMIP and nAMIP only reflect the effects of winds on precipitation variations, not all the external forcings.

Response:

You are correct, only horizontal winds are nudged toward the reanalysis data in the nAMIP experiment. And yes, the difference between AMIP and nAMIP experiments is from the nudged winds. Nudging horizontal winds can influence precipitation through their impacts on atmospheric dynamics and moisture transport, such as the positioning and strength of high and low-pressure systems. These synoptic-scale patterns can significantly impact precipitation patterns.

The AMIP and nAMIP experiments complement the iCMIP experiment. The comparison between CMIP and iCMIP experiments sheds light on the influence of internal variability in reproducing the observed wetting trend. However, the fundamental driver behind this trend—whether it originates predominantly from atmospheric internal variability or the underlying SST/Sea-ice conditions—remains ambiguous. So we first use the AMIP experiment, where oceanic conditions are presumed to be accurately predicted, to isolate the influence of atmospheric internal variability. But, the AMIP experiment fails to simulate the observed wetting trend, exhibiting almost no discernible trend. This suggests that the underlying SST/Sea-ice conditions alone are inadequate to replicate the observed wetting trend. Conversely, when the horizontal winds are nudged to observations, the nAMIP experiment shows considerable skills in replicating both the observed wetting trend ($0.21 \text{ mm d}^{-1} \text{ dec}^{-1}$; $p < 0.05$) and interannual variations ($R = 0.79$; $p < 0.05$) (Fig. 2b). These findings indicate that atmospheric internal variability is the primary driver of the wetting trend, while the impact of oceanic conditions, if any, is limited.

(6) Figure 1 should be improved. Where is the definite boundary for the TGD region. Is it only the region within the black rectangle?

Response:

Thank you for your suggestion. We have updated Fig. 1 to enhance the clarity of the precipitation contour (please see it below). In this study, the Taklamakan and Gobi Desert (TGD) region is defined by the black rectangle. This definition has now been explicitly stated in both the main text (P2L64) and relevant figure captions.

Fig. 1 | Observed summer precipitation features. a, Topographic features overlaid with long-term mean precipitation (mm d^{-1}) based on the CPC dataset (contours) and station records (dots) over the inner Eurasian continent. The subregion of Taklamakan and Gobi Desert (TGD) is delineated by the black rectangle. b, Time series of TGD-average summer precipitation anomalies (units: mm d^{-1}) based on multiple datasets during 2000–2019. The anomalies are calculated by subtracting the mean value during the overlapping period of 2000–2014 (indicated by the number in the parentheses). Linear trends of TGD-average summer precipitation (units: $\text{mm d}^{-1} \text{dec}^{-1}$) during 2000–2014 (circles) and 2000–2019 (triangles) are included. The trends are slightly shuffled horizontally for better visualization.

(7) In Figure 1. The authors discussed the different trends in 2000-2014 and 2000-2019. Why? not 2000-2009 vs. 2010-2019, or 2000-2009 vs. 2000-2019, etc. Authors should declare the reasons for the selection of different series. Additionally, the short-term trends are not credible.

Response:

Sorry for the confusion. The default AM4 and CM4 models participating in IPCC CMIP6 were originally simulated only up to the year 2014. We have now conducted additional

runs of the AM4 model, extending the simulation to 2019. However, due to computational resource limitations (CM4 includes a 1/4 degree ocean component), we were unable to extend the CM4 experiment. The updated results have been incorporated into Fig. 2 (please see below) and the main text (P13L471–472): “The CMIP simulations cover the period of 2000–2014, while AMIP, iCMIP, and nAMIP span 2000–2019.”

We acknowledge the reviewer's concern regarding the credibility of trend analysis over a 20-year period due to the limited sample size. However, our analysis provides compelling evidence supporting the robustness of the observed wetting trend: (1) The significant wetting trend observed is consistent across multiple datasets, including both gauge-based and satellite-based observations; (2) Though simulations utilizing prescribed SST (AM4) or fully coupled ocean models (CM4) fail to replicate the observed wetting trend, when these models incorporate initialized conditions, they successfully reproduce observed interannual variability and trends within the 20-year simulations. This suggests that internal variability, particularly atmospheric noise, plays a significant role in driving the wetting trend over this 20-year period; (3) The role of internal variability in the wetting trend within 20 years is further supported by the piControl simulations, where simulated precipitation responds solely to internal variability, as demonstrated through the analogue analysis. These findings, derived from both observational data and model simulations, underscore the robustness of the recent wetting trend.

Fig. 2 | Trends in summer precipitation across simulation hierarchy. a, Time series of summer precipitation anomalies (units: mm d^{-1}) averaged over Taklamakan and Gobi Desert region based (black rectangle in Fig. 1a) on CMIP and iCMIP experiments during 2000–2019. The anomalies are calculated by subtracting the mean value during the overlapping period of 2000–2014. b, Same as (a), but for AMIP and nAMIP experiments. Note, the CM4 experiment ends in the year 2014. The best linear fit and prediction errors are represented by the dashed line and shading, respectively. The long-term mean (units: mm d^{-1}), linear trend (units: $\text{mm d}^{-1} \text{dec}^{-1}$), and the correlation coefficient (units: dimensionless) with the CPC dataset are listed. Linear trends with an asterisk denote statistical significance at the 95% confidence level.

(8) In Figure S1. For i and j subplots, what is the meaning for 0101-0120?

Response:

The 0101-0120 represents a 20-year segment (year 101 to year 120) of both the piControl and 4xCO₂ simulations. The piControl simulation serves as a reference run, spanning a prolonged period (650 years in our simulations) where the model is subjected to constant pre-industrial conditions, while the 4xCO₂ experiment simulates a scenario where CO₂ concentration is quadrupled relative to pre-industrial levels. Both simulations represent theoretical scenarios of climate stability rather than real-world periods. For our analysis, we select the 20-year segment (year 101 to year 120) after these simulations have been running for 100 years, reaching a steady state. Please note, the 4xCO₂ experiment is no longer used in this study, we have removed it in the revision (please see updated Fig. S1 below).

Figure S1. Long-term mean of summer precipitation (units: mm d^{-1}) over the Taklamakan and Gobi Desert (TGD) region. a, CPC dataset, b-i, different model experiments with their name listed in the top-left corner. The TGD region is delineated by the black rectangle. The grey isoline represents the 2,000-m contour of surface elevation.

(9) In Figure S2, we can find that there are dramatically different between the stations and other datasets, especially for the significant trends. It should be more discussed in this work.

Response:

Thank you for pointing this out! We carefully re-examined the station records and found there are two primary reasons for the differences from the gridded datasets observed in the original Figure S2. First, the original station records, obtained from the China Meteorological Administration (CMA), only cover the period from 2000 to 2014, whereas the gridded dataset is from 2000 to 2019. Second, all data samples were utilized in the trend analysis without undergoing any data quality control.

In the revision, (1) we have extended the station records to the year 2019 with the latest version of the Global Historical Climatology Network monthly (GHCNm) precipitation dataset. While the total number of stations has reduced because not all stations from CMA are part of the GHCN network, the updated dataset includes stations from Kazakhstan and Mongolia, providing a broader and more uniform spatial coverage of the study area; (2) we have applied strict data quality control for the data samples. Specifically, for each station, we required a minimum of 95% availability of samples during the 2000–2019 period.

Fig. R1 demonstrates the results based on the CMA stations (covering 2000–2014) after implementing data quality control. It is evident that after excluding stations with numerous missing values, the outcomes exhibit greater alignment with the gridded datasets. Specifically, when interpolating precipitation trends from gridded datasets onto these stations, correlation coefficients exceeding 0.87 are observed across all datasets. Moreover, the updated Fig. S2, based on the GHCN datasets (covering 2000–2019), also aligns well with all the gridded datasets. These updated findings affirm the consistency of precipitation trends derived from station records with gridded datasets.

Figure R1. Linear trends of summer precipitation over TGD during 2000–2014 based on multiple precipitation datasets. Stippling in (a)-(g) indicates regions where the trends are statistically significant at the 95% confidence level. Colored circles in (a) are results based on the CMA station records. The bigger circles indicate the trends are statistically significant at the 95% confidence level. The TGD region is delineated by the black rectangle. The gray isoline represents the 2,000-m contour of surface elevation.

Figure S2. Linear trends of summer precipitation over TGD. a, CPC dataset, b-h, different model experiments. Stippling in (a)-(h) indicates regions where the trends are statistically significant at the 95% confidence level. Colored circles in (a) are results based on the station records. The bigger circles indicate the trends are statistically significant at the 95% confidence level. The TGD region is delineated by the black rectangle. The gray isoline represents the 2,000-m contour of surface elevation.

(10)Line 83-89, the further discussion should be added to make the conclusion be reliable and credible.

Response:

Thank you for your suggestion. Both external forcing and internal variability interact and can influence climate changes across various temporal and spatial scales. In the CMIP experiment, simulated precipitation results from a combination of internal variability and forced response. While the ensemble mean of CMIP experiments captures the wetting trend (though statistically insignificant), there is considerable spread within the ensemble. Therefore, the CMIP experiment does not clearly attribute the wetting trend to either

external forcing or internal variability. Consequently, in the subsequent analysis, we examine iCMIP, AMIP, and nAMIP experiments to isolate internal variability (atmospheric versus oceanic) and external forcing. We have expanded the discussion in the main text (P3L86–96):

“The simulated precipitation in the CMIP experiment can be viewed as a combination of internal variability and forced response. External forcing typically operates over longer time scales due to gradual or persistent changes over time. Internal variability tends to dominate over relatively shorter time scales, driven by internal processes and feedback mechanisms within the climate system that can vary rapidly and chaotically. Notably, internal variability-induced changes can contribute significantly to climate changes, and even surpass externally-forced changes on local to continental scales, particularly in mid- and high-latitude regions^{25–27}. This influence may persist over several decades. The large spread across the three CMIP ensemble members suggests the possibility of internal variability playing an important role in shaping the observed wetting trend.”

Reviewer #2 (Remarks to the Author):

This paper uses a series of GFDL model simulations to argue that recent observed trends of increasing precipitation over deserts in interior Asia are due to internal variability, not external forcing such as greenhouse gas emissions. I found the paper analyses compelling and thorough, the writing clear, and the figures helpful. The insight into the trends of this region is relevant for environmental management in arid Asia, and the methodology for distinguishing whether trends are due to internal variability can be usefully applied to other regions in the future. I think this paper will be appropriate for acceptance to Nature Communications following some minor revisions recommended below. Most of these comments are small errors or notes to clarify certain methodologies better. One comment bolded below is more substantial and pertains to the need for further discussion of how reliable (or not) the observed wetting trend is.

Response:

We very much appreciate the reviewer's insightful and constructive comments and suggestions on our manuscript. Please see our point-by-point response below.

1. [44-45] To clarify does “warm-dry” vs “warm-wet” refer to a climate that is like that in the annual mean, or does it mean that the warm season (eg boreal summer) is shifting from dry to wet? If this sentence could be revised slightly to make this clearer that would be helpful.

Response:

Good point! The cited references refer to both summer and annual mean temperature and precipitation. The conclusion persists when we assess their changes using the datasets employed in our analysis (Fig. R2). We have revised the text to reflect these changes (P2L43–45):

“On one hand, more precipitation provides much-needed relief from drought conditions in this region, potentially shifting its climate from a warm-dry to a warm-wet regime, both in summer and annually^{9–11”}.

Figure R2. a. Time series of summer and annual precipitation anomalies (units: mm d⁻¹) averaged over the Taklamakan and Gobi Desert region based on the CPC precipitation dataset during 1980–2019. b. Same as (a), but for surface temperature based on ERA5 reanalysis. The anomalies are calculated by subtracting the mean value during the entire period. The best linear fit and prediction errors are represented by the dashed line and shading, respectively. Linear trends are listed for both summer and annual time series.

2. [70] Stations are not shown in Fig. S1a. Do you mean Fig. 1a which does have stations?
Corrected.

3. [73-75,S2a] In the places where the station records are drying they often look at odds with the gridded dataset according to S2a. The paper seems to be premised on the gridded record being more reliable, since it is focused on pronounced wetting trends. I think some discussion should be added regarding whether the station records or gridded products are more reliable in this region. Do we trust gridded products' ability to pick up a trend in this region when there aren't many stations? Relatedly, Fig S2a indicates that the trends tend to be more significant over mountains, and only sporadically significant over the deserts.

Response:

This is a great question that frequently arises when analyzing precipitation in regions with limited station observations. We acknowledge potential issues in our original analysis, where all the data samples were used without any data quality control, and results from station records covering 2000–2014 were directly compared with the gridded datasets spanning 2000–2019. In this revision, we have made the following two modifications:

- (1) We have carefully examined the precipitation trends from station records during 2000–2014, requiring at least 95% available samples for each station. We then compared the results with the other gridded precipitation for the same period.
- (2) We have extended the station records to 2019 and updated the results discussed in this study.

Based on these new analyses, we found better consistency between station records and gridded precipitation datasets after removing stations with numerous missing values for both periods. The trend based on the updated station records during 2000–2019, covering a broader spatial coverage of our study region, shows comparable results with the CPC dataset (please see updated Fig. S2 below). For more details, please refer to our response to comment #9 from the 1st reviewer.

Figure S2. Linear trends of summer precipitation over TGD. a, CPC dataset, b-h, different model experiments. Stippling in (a)-(h) indicates regions where the trends are statistically significant at the 95% confidence level. Colored circles in (a) are results based on the station records. The bigger circles indicate the trends are statistically significant at the 95% confidence level. The TGD region is delineated by the black rectangle. The gray isoline represents the 2,000-m contour of surface elevation.

4. [100-101,409-411] Can you please clarify here or in the methods why the ensemble analogue is based on 10 not 20 year segments? That wasn't obvious to me especially given that the observed record considered is closer to 20 years.

Response:

The 10-year segment is selected to accommodate the relatively short piControl simulation used in this study, which spans 650 years. If a 20-year segment were used with the same correlation coefficient threshold of 0.6, it would restrict the ensemble members to 12. However, relaxing the correlation coefficient to 0.4 allows for the inclusion of 35 members.

Nonetheless, the results obtained from the 20-year segment remain consistent with those derived from the 10-year or 5-year segment (Fig. R3).

Figure R3. Composite precipitation trend based on variability analogue ensembles. a, Linear trends of summer precipitation based on the variability analogue ensembles using 20-year segments. The TGD region is delineated by the black rectangle. The gray isoline represents the 2,000-m contour of surface elevation. b, Time series of the summer precipitation averaged over the TGD region based on the variability analogue ensembles (blue) and CPC dataset (black). The light blue shading denotes the spread among the respective ensembles. The mean, linear trend, and correlation coefficient with the CPC dataset are shown for variability analogue ensembles using 20-year segments.

5. [111] I think you mean S2 which has trends. S1 has means.
Corrected.

6. [111-116] I think you should specify somewhere in this sentence that you are referring to what you find in TGD, otherwise it seems like an oddly general statement.
Response:

Thank you for your suggestion. We have revised these sentences to clarify this, now it reads (P4L133–138):

“This finding demonstrates that when appropriately initialized with specific atmospheric and oceanic states, a dynamic model like the one employed in this study has the capability to predict the time evolution of the weather/climate system (i.e., internal variability) in the TGD region within a defined limit, beyond which memory loss occurs due to a combination of error amplification and model limitations.”

7. [151] Suggest to cut “Especially,”.

Corrected.

8. [199-200] I suggest to soften this statement and perhaps more explicitly acknowledge the biases—it doesn’t appear to me that “all” the synoptic features are simulated reasonably. In particular, iCMIP moisture fluxes appear to be in opposite direction in east part of box for cluster.

Response:

Thank you for your suggestion. We have revised this sentence to (P7L232–238):

“The nAMIP and iCMIP simulations capture the general synoptic features, but notable biases are evident when compared to observations. Based on the geopotential height anomaly, the slightly poleward-shifted wave pattern in the second cluster is distorted in the nAMIP simulation, while the zonal wave pattern in the third cluster is misrepresented in the iCMIP simulation. Similar discrepancies are observed in the low-level moisture fluxes. These biases may be partly attributed to the smaller sample size in the model simulation, warranting further investigation.”

3. Geopotential height anomalies look somewhat different too.

Response:

Please see our response to your comment above.

9. [273-274] This archived manuscript (presumably under review?) seems to have substantial overlaps in the author team and subject matter. For the record, could you please clarify the distinction in subject matter or methodology? I assume they are distinct enough to be published separately. I just want to double check.

Response:

Thank you for bringing this up. In this separate manuscript, we applied the clustering analysis to the CPC precipitation products over a longer period (1979–2019). We aimed to elucidate the underlying large-scale patterns for each precipitation cluster, thus providing a mechanistic understanding of the observed wetting trend over the TGD region. This analysis focused on two primary objectives: (1) Detailed analysis of precipitation clusters. The examination of the spatial pattern of the identified precipitation clusters revealed distinct characteristics, with the second cluster (C2) associated with intense precipitation over the western region of the TGD region, and the third cluster (C3) over the eastern part. By analyzing the evolution of large-scale circulation anomalies conditioning on different precipitation clusters, we found these intense precipitation events in C2 and C3 are triggered by upper-tropospheric disturbances in the form of transient Rossby wave packets (see Fig. R4).

Figure R4. Upper-tropospheric disturbances. Vertical structures of the composite anomalies in geopotential height averaged over 30°–50°N on Day -2 (top row), 0 (middle row) and 2 (bottom row) in C2 (left column) and C3 (right column). Vertical dash lines are the longitudinal boundaries for the TGD region.

(2) Detailed statistics of precipitation clusters. By analyzing precipitation time series from different clusters, we revealed that the wetting trend is primarily attributable to C3, with a lesser influence from C2 (as shown in Fig. R5). This observation aligns with the more pronounced wetting trend observed over the eastern region of TGD. Moreover, we further analyzed the frequency, duration, and intensity of precipitation events in each cluster. We found the frequency and duration sometimes cancel out each other or demonstrate moderate changes, while the increase in precipitation intensity plays a larger role in the wetting trend.

Figure R5. Contribution of precipitation frequency, duration, and intensity. Time series of the (A) combined precipitation, (B) frequency, (C) duration, and (D) intensity of the three clusters. For each series, the best linear fit and prediction errors are represented by the dashed line and shading, respectively. The linear trends are given, with asterisks denoting statistical significance at the 95% confidence level.

Since the three precipitation clusters have proven useful in comprehending the recent wetting trend, this separate study serves as a valuable reference for our present analysis. The robustness of our hierarchical model simulations relies on their ability to accurately capture the observed clustering analysis. This validation is demonstrated in Fig. 5, adding greater confidence in the fidelity of our model simulations.

10. [337-342] I was a bit surprised APHRODITE was not included among the datasets you analyzed since it focuses on high-resolution Asian precipitation. Could you try adding this dataset or explain why it is insufficiently distinct to be needed?

Response:

Thank you for raising this point. While the APHRODITE (abbreviated as APHRO) data initially wasn't on our radar as we focused on the relationship between summer NAO, storm activity, and precipitation across Eurasia, we recognize its value for cross-validation purposes, especially since it incorporates data from a comprehensive rain gauge observation network across Asia.

In our revision, we have incorporated the latest version (V1901) of the APHRO dataset (2000–2015) into the comparison (see updated Fig. 1 below). When compared with other precipitation datasets, the TGD-averaged precipitation from APHRO shows a slightly lower value (0.98 mm d⁻¹), yet it exhibits high intercorrelation efficiencies with them during their respective overlapping periods ($R > 0.94$). The linear trend derived from APHRO during 2000–2014 ranks third among all 8 datasets, smaller than CPC and GPCP

datasets. We have made updates to the Methods section to reflect these changes (P11L376–388):

“Multiple widely used gridded monthly precipitation datasets, including NOAA/Climate Prediction Center (CPC) precipitation dataset⁴¹, U of Delaware (UDeI) dataset⁴², Tropical Rainfall Measuring Mission (TRMM) 3B43⁴³, CPC Merged Analysis of Precipitation (CMAP) dataset⁴⁴, NOAA's Precipitation Reconstruction over Land (PREC/L) dataset⁴⁵, Global Precipitation Climatology Project (GPCP) dataset⁴⁶, and Asian Precipitation-Highly-Resolved Observational Data Integration Towards Evaluation (APHRODITE; shorted for APHRO) dataset⁴⁷ are compared in this study to examine the wetting trend, among which the NOAA/Climate Prediction Center (CPC) precipitation dataset is adopted as the main one since the daily coverage makes it uniquely suitable for the subsequent clustering analysis (to be detailed). Precipitation measurements from 47 observation stations located in Northwest China, Kazakhstan, and Mongolia, retrieved from the Global Historical Climatology Network monthly (GHCNm) precipitation dataset, are utilized as a reference (see Fig. 1a for their geographical locations).”

Fig. 1 | Observed summer precipitation features. a, Topographic features overlaid with long-term mean precipitation (mm d^{-1}) based on the CPC dataset (contours) and station records (dots) over the inner Eurasian continent. The subregion of Taklamakan and Gobi Desert (TGD) is delineated by the black rectangle. b, Time series of TGD-average

summer precipitation anomalies (units: mm d⁻¹) based on multiple datasets during 2000–2019. The anomalies are calculated by subtracting the mean value during the overlapping period of 2000–2014 (indicated by the number in the parentheses). Linear trends of TGD-average summer precipitation (units: mm d⁻¹ dec⁻¹) during 2000–2014 (circles) and 2000–2019 (triangles) are included. The trends are slightly shuffled horizontally for better visualization.

11. [342-344] Please specify where the station-based measurements were retrieved from.
Response:

The station-based measurements were retrieved from the Global Historical Climatology Network monthly (GHCNm) precipitation dataset (<https://www.ncei.noaa.gov/data/ghcnm/v4beta/>). We have included the details in the Method section (P11L385–388).

12. [345-348] Please specify a bit more detail about how the NAO index was derived—maybe the observed dataset it was derived from and a citation for the methodology?
Response:

We utilized the Hurrell principal component (PC)-based North Atlantic Oscillation (NAO) index, as introduced by Hurrell et al. in their Geophysical Monograph (2003). This index is derived through empirical orthogonal function (EOF) analysis, which decomposes sea level pressure anomalies into a series of uncorrelated and orthogonal patterns. The primary EOF pattern, characterized by a meridional dipole between Iceland/Greenland and the Azores, represents the NAO pattern. The associated time series of the principal component (PC), obtained by projecting the EOF onto the anomaly field, represents the NAO index. We have added the reference and elaborated on this method in the main text (P11L391–396):

“This index is derived through empirical orthogonal function (EOF) analysis, which decomposes sea level pressure anomalies over the Atlantic sector (20°–80°N, 90°W–40°E) into a series of uncorrelated and orthogonal patterns. The primary EOF pattern, characterized by a meridional dipole between Iceland/Greenland and the Azores, represents the NAO pattern. The associated time series of the principal component (PC), obtained by projecting the EOF onto the anomaly field, represents the NAO index.”

Reference:

Hurrell et al. (2003) in The North Atlantic Oscillation: Climate Significance and Environmental Impact, 2003. J.W. Hurrell, Y. Kushnir, G. Ottersen, and M. Visbeck, Eds. Geophysical Monograph Series, 134, 279pp.

13. [353-355] I suggest to rephrase this sentence slightly, since SPEAR is run at a few different horizontal resolutions, but as currently phrased it makes it seem like it is always ~100 km. Are all the simulations you analyze run at ~100 km?

Response:

You're right. All simulations are conducted at approximately 100 km resolution. While results from SPEAR-MED exhibit similar findings, we opted to utilize the SPEAR-LO

results in this study to mitigate any potential impacts stemming from horizontal resolution differences. We have revised this sentence to provide clarification on this (P12L412–415):

“Note, SPEAR can run with different atmospheric horizontal resolutions ranging from 1° to 0.25°. We opt for the 100-km configuration (i.e., SPEAR-LO) to ensure consistency with other simulations, thus mitigating any potential impacts arising from differences in horizontal resolution.”

14. [359-361] It would be better if this list focused on extreme weather events that were relevant to the TGD region, or at least noted which ones were relevant to this region.

Response:

This is a valid point. We haven't used these models to specifically study this region before. But simulations from AM4 and CM4, as well as other generations of GFDL models, have been extensively used in studying hydroclimate dynamics in similar regions (In this regard, please refer to our response to comment #2 from the 1st reviewer for more details).

Moreover, following the suggestion from the 1st reviewer, we have now incorporated precipitation data from 40 GCMs participating in CMIP6. We have compared their simulation of precipitation mean features over the TGD region with the GFDL models analyzed in this study. Notably, the results reveal that the GFDL models perform well in simulating both the mean precipitation and its spatial distribution compared to the other CMIP6 GCMs. We have incorporated this comparison into the main text for completeness (P3L78–83):

“These wet biases are common to CMIP5²⁴ and CMIP6 models²³, and might be related to the representation of topographic features over this region. Nevertheless, in terms of long-term mean and spatial distribution, results from GFDL model simulations outperform the majority of the CMIP6 models (Fig. S3), adding confidence in their suitability for comprehending precipitation variability in the TGD region.”

Figure S3. a, Long-term mean of summer precipitation (units: mm d⁻¹) based on CMIP6 multi-model mean. The TGD region is delineated by the black rectangle. The gray isoline represents the 2,000-m contour of surface elevation. b, Boxplot of mean bias of summer precipitation averaged over the TGD region based on 40 CMIP6 models. Results from GFDL models used in this study are denoted by green circles and are slightly shuffled horizontally for better visualization. c, Similar to (b), but for the precipitation pattern correlation calculated over the TGD region. Mean bias and pattern correlation are calculated relative to the CPC precipitation dataset.

15. [365-367] Just double checking that the iCMIP simulations with SPEAR are run at 100 km resolution (eg is it SPEAR-LO or SPEAR-MED)?

Response:

The iCMIP simulation is based on SPEAR-LO, which operates at a 100-km resolution. We have specified this in the main text (P12L412–415).

16. [379-381] What is the timescale of the nudging? Also can you please clarify more specifically in the text what it means that only the large-scale winds are nudged?

Response:

Nudging is a data assimilation method that introduces an additional term in the model equations to drive the model towards a reference state, which, in our study, is based on the MERRA2 reanalysis (Gelaro et al., 2017). In the nAMIP experiment, the model's horizontal winds at each level are nudged to the 3-hourly averaged products from the MERRA-2 reanalysis. The nudging timescale in the nAMIP simulation is set to 6 hours.

However, other variables, such as temperature and humidity, are not nudged in this study. This decision is made to achieve a balance between model dynamics/physics and observational constraint, as incorporating additional nudging variables can sometimes lead to significant inconsistencies in the mean climate. Figure R6 shows the simulated precipitation during the 2005 hurricane season using different nudging strategies. Notably, the experiment with further nudging temperature and humidity produces a larger positive precipitation bias compared to the one only nudging horizontal winds. Similar discrepancies are observed for individual extreme precipitation events. This is because nudging sometimes can potentially override or suppress the dynamics and physics of the model, especially in the AMIP-type simulation when prescribed SST is used. This can lead to strange or unrealistic results because the model is being forced to conform too closely to the observed data, rather than allowing the inherent dynamics of the system to play out. Similar findings have been observed in other GCMs. For instance, Sun et al. (2019) conducted experiments with different combinations of nudging variables in the Energy Exascale Earth System Model (E3SM). They discovered that nudging temperature or specific humidity in addition to horizontal winds could enhance spatial and temporal anomaly correlations of temperature and humidity, but resulted in notable changes in their long-term climatology. To keep the model's fidelity to the underlying physical processes, we only nudge the horizontal winds in our study. We have added additional details about the nudging set up in the main text (P13L460–469):

“In the fourth type, the nudged AMIP simulation (nAMIP), large-scale winds are nudged to reanalysis data, rather than allowing the atmospheric model to generate its own winds. This nudging setup is similar to that used by an earlier study⁶⁶, but only large-scale winds are nudged in nAMIP. It's important to note that specific humidity, clouds, and precipitation are computed interactively in both AMIP and nAMIP simulations, subject to the same dynamic and physical processes as in the AMIP setup. Other variables, such as temperature and humidity, are not nudged in this study. This decision is made to achieve a balance between model dynamics/physics and observational constraint, as incorporating additional nudging variables can sometimes lead to significant inconsistencies in the mean climate.”

Averaged precipitation during 2005 Hurricane Season

Figure R6. Global mean of precipitation (units: mm d^{-1}) during the 2005 hurricane season (June to November) based on (a) observation, (b) AM4 with nudged horizontal winds, (c) AM4 with nudged horizontal winds, temperature, and humidity. Global mean values are listed in the top-right corner.

Reference:

Gelaro, R., McCarty, W., Suárez, M. J., Todling, R., Molod, A., Takacs, L., ... & Zhao, B. (2017). The modern-era retrospective analysis for research and applications, version 2 (MERRA-2). *Journal of Climate*, 30(14), 5419-5454.

Sun, J., Zhang, K., Wan, H., Ma, P. L., Tang, Q., & Zhang, S. (2019). Impact of nudging strategy on the climate representativeness and hindcast skill of constrained EAMv1 simulations. *Journal of Advances in Modeling Earth Systems*, 11(12), 3911-3933.

17. [396] It's a bit unclear to me what aspects of the clusters is assessed to be similar or not? Is it the spatial pattern of precipitation?

Response:

We employ Ward's minimum variance method for the clustering analysis (Ward 1963; Zhao et al. 2017). At each step, this method uses the Euclidean distance between each pair of clusters during the merging process. Consequently, the similarity within each precipitation cluster accounts for both its spatial pattern and the absolute values. To clarify, we have revised this sentence (P13L482–484):

“The similarity between each single-member cluster is calculated using Ward’s minimum variance method, which measures the Euclidean distance between each pair of clusters.”

Reference:

Ward Jr, J. H. (1963). *Hierarchical grouping to optimize an objective function*. *Journal of the American statistical association*, 58(301), 236-244.

Zhao, S., Deng, Y., & Black, R. X. (2017). *A dynamical and statistical characterization of US extreme precipitation events and their associated large-scale meteorological patterns*. *Journal of Climate*, 30(4), 1307-1326.

18. [Fig 1a] The contours of precip are hard to see. Can you make them a bit thicker?
Done. Please see the updated Fig. 1 in our response to your comment #10.

19. [Fig 1b] To clarify the time series is just the average in the box, right? No additional masking for dry regions is done?

Response:

That's correct. The time series represents the average within the specified box without any additional masking applied.

20. [589] Do you mean "triangles"? I don't see "squares".

Corrected.

21. [593] I suggest to call the start of this caption "Trends in summer precipitation across simulation hierarchy" instead.

Done.

22. [596, 610] I think "substrating" should be "subtracting".

Response:

Thank you for catching those typos! We have corrected them.

23. [Fig 3c] It's not obvious to me why the pattern correlation between the analogue ensemble and the observed data should be high, given that as far as I understand the analogue ensemble is based on area average time series. Am I misunderstanding something and the analogue ensemble includes a spatial component?

Response:

The analogue ensemble relies solely on area-averaged time series and does not incorporate spatial patterns. The pattern correlation between individual members from the analogue ensemble and the observations ranges from 0.10 to 0.67. The higher correlation observed in the ensemble mean compared to individual ensemble members can be attributed to the averaging of multiple members. This averaging process results in a reduction of random errors and an amplification of the signal-to-noise ratio.

24. [614-615] Please clarify in the caption what the dashed vs line contours means.

Response:

Thank you for pointing this out. We have clarified in the caption that dashed contours represent negative trends, while solid contours indicate positive trends (P22L740–742).

25. [615-616] You mean that the stippling is where the analogue ensemble linear trends are significant correct?

Response:

Yes, we have changed “dots” to “stippling” in the caption throughout the main text and supplementary materials where applicable.

26. [631] Suggest to call the start of this caption “Cluster analysis of summer precipitation across simulation hierarchy” instead.

Corrected.

27. [637-638] I think this note about the gray isoline referring to topography needs to be added to other figure captions that contour topography too.

Added.

28. [SI 20] should say “CPC dataset” not “CPC datasets” since it is one dataset

Corrected.

29. [SI 23] again should say “CPC dataset”

Corrected.

30. [Fig S6] It's weird that precip in mm/day just shows up as xlabel in middle panel. suggest to have at bottom instead or all panels? also what is the y axis unit?

Response:

Thank you for pointing this out. We have moved the x-label to the bottom panel and have added the units $(\text{mm day}^{-1})^{-1}$ for the y-axis.

31. [Fig S7] Is the black box TGD? Please note in the caption.

Added.

REVIEWERS' COMMENTS

Reviewer #1 (Remarks to the Author):

No any more comments for this revised manuscript. I think this revised manuscript will be appropriate for acceptance to Nature Communications.

Reviewer #2 (Remarks to the Author):

I sincerely thank the authors for their thorough and clear response to the reviewer comments. The further work that resolves the discrepancies in station-based vs gridded dataset precipitation trends over TGD strengthens the manuscript. I recommend the manuscript be accepted, but have one tiny suggestion below:

Regarding your answer about 10 vs 20 year segments for the analogue trend analysis, I suggest a slight wording change in the manuscript so that hopefully this isn't a source of confusion for future readers. Perhaps you could simply add to line 445 after "10-year segments" the words in parentheses "(as opposed to 20-year segments)".

We express our gratitude to the editor and reviewers for their valuable feedback and the time they dedicated to carefully re-reviewing our manuscript, and are happy that the reviewers found our changes to be satisfactory. Please find our point-by-point responses below to the second reviewer's suggestion.

REVIEWER COMMENTS

Reviewer #1 (Remarks to the Author):

No any more comments for this revised manuscript. I think this revised manuscript will be appropriate for acceptance to Nature Communications.

Reviewer #2 (Remarks to the Author):

I sincerely thank the authors for their thorough and clear response to the reviewer comments. The further work that resolves the discrepancies in station-based vs gridded dataset precipitation trends over TGD strengthens the manuscript. I recommend the manuscript be accepted, but have one tiny suggestion below:

Regarding your answer about 10 vs 20 year segments for the analogue trend analysis, I suggest a slight wording change in the manuscript so that hopefully this isn't a source of confusion for future readers. Perhaps you could simply add to line 445 after "10-year segments" the words in parentheses "(as opposed to 20-year segments)".

Response:

Good point, we have add this to the sentence, and now it reads (P11L343–345): *“The selection of 10-year segments (as opposed to a 20-year segment) is justified in the sense that any linear trends due to decadal variability can be largely preserved while also accommodating the relatively short piControl simulation.”*